# Association between Changes in the Regularity of Working Hours and Cognitive Impairment in Middle-Aged and Older Korean Workers: The Korean Longitudinal Study of Aging, 2008–2018

**DOI:** 10.3390/ijerph19074161

**Published:** 2022-03-31

**Authors:** Won-Tae Lee, Sung-Shil Lim, Jin-Ha Yoon, Jong-Uk Won

**Affiliations:** 1Department of Occupational and Environmental Medicine, Severance Hospital, College of Medicine, Yonsei University, Seoul 03722, Korea; lewot20@yuhs.ac (W.-T.L.); lssmail@yuhs.ac (S.-S.L.); 2The Institute for Occupational Health, College of Medicine, Yonsei University, Seoul 03722, Korea; flyinyou@yuhs.ac; 3Department of Public Health, Graduate School, Yonsei University, Seoul 03722, Korea; 4Department of Preventive Medicine, College of Medicine, Yonsei University, Seoul 03722, Korea

**Keywords:** work schedule tolerance, mental status, dementia tests, workload, cognitive dysfunction, middle-aged and older population

## Abstract

Changes in the regularity of working hours affect the cognitive function of middle-aged workers. This study investigated the association between alterations in the regularity of working hours and cognitive impairment in middle-aged and elderly Korean workers. The data from the Korean Longitudinal Study of Aging were analyzed and cognitive function was evaluated using the Korean version of the mini–mental state examination. A score of <23 points was defined as cognitive impairment. The effect of changes in the regularity of working hours on cognitive impairment development was assessed using the generalized estimating equation model. Compared with regular working hours group, the odds ratios (ORs; 95% confidence interval) of the “consistently irregular”, “regular to irregular”, and “irregular to regular” groups were 1.56 (1.30–1.88), 1.46 (1.20–1.77), and 1.24 (1.01–1.53), respectively. The risk of cognitive deterioration was found in the “consistently irregular” group. However, only workers with normal working hours in the “regular to irregular” group had a significant risk of cognitive deterioration (1.51 (1.21–1.89)). Altered working hours were associated with cognitive impairment in middle-aged and older workers. The study emphasizes the need to implement a standard work schedule that is suitable for middle-aged workers.

## 1. Introduction

Working environments include a variety of work schedules accompanied by occupational diversity. Nonstandard work schedules, such as irregular work hours, shift work, and work with flexible start times cause a wide range of physical and mental outcomes [1]. Shift work can cause socioeconomic problems along with various health problems such as, injuries, cardiovascular diseases, digestive system diseases, cancer, and neurological diseases [2,3]. Atypical work schedules affect circadian rhythm, lead to poor sleep quality, and affect mental health, thereby affecting the workers’ wellbeing [4,5,6].

Cognitive impairment is a public health catastrophe without ideal treatment. The impairment is common in the elderly population, and the resulting burden of healthcare costs is high [7,8]. The associations of occupation or work environment with decline in cognitive functions remain unclear, although numerous studies have been conducted to investigate this association [9]. In addition, early diagnosis and prevention of increased cognitive vulnerability are important as symptoms develop over time and are mild rather than acute [10].

Korea is a rapidly aging country, with a large population aged ≥50 years. In 2020, the elderly population in Korea (aged more than 65 years) accounted for 16.4% of the total population [11]. Additionally, since the economically productive population among middle-aged people is increasing, the industrial productivity in this age group is economically important in Korea. However, their working environment and its effects on health are also concerning. Studies have analyzed the link between work environment and cognitive functions. According to a longitudinal study of middle-aged people in the UK, workers who worked for more than 55 h a week had lower cognitive ability when measured at intervals of 5 years through various cognitive ability tests [12]. A systematic review showed that cognitive decline and dementia increased as the job complexity increased [13].

With the work schedules and working environment of middle-aged and older workers becoming important issues, their cognitive health effects can suggest better policies and guidelines for them. However, there are limited studies on the association between irregularity of working hours and cognitive function, and the results that are reported, are controversial. Given these diverse findings, we carried out a longitudinal analysis of changes in the regularity of working hours and cognitive impairment by controlling confounders that were not fully controlled in previous studies. Our study thus analyzed whether cognitive decline occurs more frequently in middle-aged and elderly workers due to changes in work regularity and the association of cognitive decline and working hours.

## 2. Materials and Methods

### 2.1. Data and Study Population

This study used data from the Korean Longitudinal Study of Aging. This study on aging was implemented by the Korea employment information service (KEIS) and received funding from the Ministry of Employment and Labor. In 2006, the KEIS extracted the data through systematic sampling, using a multistage design of stratification by region and residential type, and provided socioeconomic and health data of middle-aged people aged ≥45 years residing in Korea. The data was collected through the computer-assisted personal interviewing (CAPI) method. Our study used data for the years 2006, 2008, 2010, 2012, 2014, 2016, and 2018. By assigning weights as suggested in the data, we ensured that our study data were representative of the Korean middle-aged and elderly population. We included data on 3826 workers who had complete information on work regularity and cognitive impairment every 2 years, from 2006 to 2018 (Figure 1).

### 2.2. Variables

The Korean version of the mini–mental state examination (K-MMSE), a screening tool for dementia, was first implemented and validated to determine the cognitive status [14]. The K-MMSE consists of 30 points derived through 19 items in seven categories: orientation for time and for place, registration, attention and calculation, recall, language, and visual construction. Since the MMSE score varies according to age and educational level, there are several normative studies that analyze each cutoff value [15,16]. However, MMSE is not a diagnostic test, but a screening function that enables disease suspension, and there is a domestic analysis suggesting that it is reasonable to set the cut-off of the MMSE for the age of the subject in our study in a traditional way [17]. Traditionally, a score of ≤17 indicates severe cognitive impairment, and a score of ≤23 indicates mild cognitive impairment [18,19]. For our study, a score of ≤23 was classified as cognitive impairment.

For variables on regularity of working hours, the answers to the following questions were used for wage workers and self-employed individuals: “Do you currently work fixed working hours?” This question was asked every 2 years, and we detected the changes in regularity by applying a 2-year lag-time to it. From the second follow-up, we categorized the changes into “consistently irregular group”, “consistently regular group”, “regular to irregular group”, and “irregular to regular group” through the gap of the 2 years.

Other socioeconomic variables and health-related covariates were also included in the study. Age was classified into four groups: 45–54, 55–64, 65–74, and ≥75 years; their residences were categorized into rural, urban, and metropolitan areas. Education was grouped as below elementary school, middle and high school, and university graduates and above, whereas household income was divided into low, low-to-mid, mid-to-high, and high income categories. Working hours were classified according to whether they worked more than 52 h per week. This classification is based on the Korean Labor Standards Act stipulating that the maximum statutory working hours per week is 52 h. In addition, there have been studies showing that chronic diseases (particularly diabetes, hypertension, and chronic kidney disease) are associated with dementia or mild cognitive impairment in the elderly [20,21,22]. Health behaviors, including tobacco and alcohol use, and the presence of other chronic diseases were expressed as health conditions. Those included in the presence of chronic diseases consist of hypertension, diabetes, malignancy, chronic lung disease, liver disease, heart disease, and cerebrovascular disease. The data on social activity were also included to indicate whether individuals were involved in social gatherings, volunteers, cultural, sports organizations, or religious gatherings. All covariates were all included in the multivariable model in the analysis.

### 2.3. Statistical Analysis

Descriptive statistics were used to describe various characteristics of the study participants and a chi-square test was used to determine significant differences between the cognitive impairment groups. The generalized estimating equation (GEE) model was used for analyzing correlated data longitudinally, to study the effect of regularity of working hours on cognitive impairment. The GEE model is mainly used to estimate the causal model of repeatedly measured panel data, and it is an analysis technique that can assess data without multiple assumptions by applying the generalized linear model of multivariate variables. Our study used this model since the health status of an individual worker could be time-dependent or dependent on other covariates that were included. A subgroup analysis was conducted based on working hours. The impact of working hours was also analyzed as the regularity of working hours. A *p*-value ≤ 0.05 was considered significant. Analyses were performed using R (The R Foundation for Statistical Computing, Vienna, Austria), version 3.6.

## 3. Results

### 3.1. Descriptive Statistics

The socio-demographic characteristics and K-MMSE scores of the study participants are shown in Table 1. Of the 3826 individuals, 465 (12.1%) had cognitive impairment. The K-MMSE score of the group without cognitive impairment was 28.2 (standard deviation [SD] = 3.54) and that of the group with low cognitive impairment was 19.9 (SD = 3.54). With reference to the regularity of working hours, 47.0% (*n* = 1797) of the participants were irregular at baseline (year 2006). Data indicated that 44.6% (*n* = 1499) in the noncognitive impairment group reported irregular work hours, and 64.1% (*n* = 298) in the cognitive impairment group reported irregular work hours. Analyzing the 2-year changes indicated that 31.8% (*n* = 1215) of the participants remained irregular, 13.9% (*n* = 531) changed from regular to irregular work hours, 15.2% (*n* = 582) changed from irregular to regular work hours, and 39.2% (*n* = 1498) reported with regular work hours. In addition to regularity, 30.1% (*n* = 1153) of the participants worked more than 52 h per week, which meant that these individuals worked more than the legal weekly working hours in Korea.

### 3.2. The GEE Model Analysis

The results of the GEE model to evaluate the changes in the regularity of working hours and the level of cognitive function are shown in Table 2. Using regularly maintained work hours groups as a reference, participants in the “consistently irregular”, “regular to irregular”, and “irregular to regular” groups had a higher OR of cognitive decline after adjusting for covariates. The “consistently irregular” group reported an OR of 1.56 (95% confidence interval (CI): 1.30–1.88), the “regular to irregular” group had an OR of 1.46 (95% CI: 1.20–1.77), and the “irregular to regular” group had an OR of 1.24 (95% CI: 1.01–1.53). Upon classification by household income and education, significant risk was observed in the low-income groups (OR 1.36, 95% CI: 1.07–1.73) and among those who received only primary education (OR 4.17, 95% CI: 2.96–5.88). The risk of cognitive impairment was not significant in the group that worked for more than 52 h but was high among socially inactive participants.

Figure 2 shows the results of subgroup analysis according to working hours during the week. The risk of cognitive deterioration was observed in the “consistently irregular” group, regardless of the working hours. Among the participants with long working hours, the “regular to irregular” group had an OR of 1.46, but it was not significant (95% CI: 0.98–2.17). However, in the group with normal working hours, these odds were significant (1.51, (95% CI: 1.21–1.89)).

## 4. Discussion

In this study, we investigated whether changes in the regularity of working hours are associated with cognitive function of the middle-aged population. Our findings confirmed that irregularity in working hours was correlated with the risk of cognitive decline. This effect was robust even when a number of possible confounding factors such as sex, education, household income, and social activity were adjusted. Moreover, there was a difference within the groups stratified according to working hours. In the group working for more than the statutory working hours, participants who maintained irregularity had a greater risk of cognitive decline than did the group with normal working hours. Supplementarily, “regular to irregular” subgroup workers had large ORs values regardless of working hours, but was only significant in the group with normal working hours.

Previous studies have shown that a work schedule does affect the physical and mental health of workers [5,23]. A longitudinal study by Marquie et al. revealed that shift work influenced impaired cognition and that 10 years of shift work resulted in approximately 6.5 years of age-related cognitive decline [24,25]. Moreover, atypical work schedules can reduce memory and the processing speeds on the next day [26]. However, a Swedish cohort study revealed that shift work among middle-aged adults was not correlated with cognitive ability at retirement [27] and the association observed between shift work history and average cognitive function among older women was weak as reported in the Nurses’ Health Study [28]. In this way, most of the previous studies have conducted an analysis of cognitive function, focusing on shift work and long working hours. Our study, in contrast, showed that the detailed risk of the regularity of working hours cannot be ignored.

Regarding the confounders that may be related to cognitive function, similar earlier studies reported that the education level affects cognitive function at the time of retirement [27]; our results also showed that the OR was 4.17 in those with an education level below elementary and was high in those educated up to middle and high school. A cohort study conducted in southwestern France found that factors such as increased social, physical, and intellectual participation were associated with a reduced risk of dementia in the overall population [29]. There have been reports that the absence of other chronic diseases, such as hypertension, hyperlipidemia, and constrictive heart failure, reduces the risk of cognitive impairment and the hazard ratio of dementia [30,31,32]. Accordingly, this study adopted an intra-individual approach to evaluate associations while adjusting for social participation, educational differences, and chronic diseases.

In the case of working hours analyzed by subgroup, the working hours were found to affect the deterioration of cognitive function. A study on middle-aged people using the same data found that women with long working hours had less cognitive ability after 5 years [33]. We expected that the effect of working hour regularity would be greater in one of the subgroups of working hours. In the “consistently irregular” group, workers with long working hours had a higher risk of cognitive decline, but in the “regular to irregular” group, only workers with statutory working hours were significant. It has been reported that regularity decreases with the increase in working hours [34]. The working environment of individuals who work for long hours may have a greater impact than the irregularity of working hours. Our results may be the result of selection bias caused by changes in the industry, move from the workplace, or changes in tasks at the workplace. However, difference in tendency between cognitive impairment among the subgroups was not observed. It was consistent regardless of the difference in working hours; the risk of cognitive decline among workers with consistently irregular working hours and those who transitioned from “regular to irregular” working hours.

The irregularity of working hours can induce circadian misalignment and various health problems due to sleep deprivation and maladaptation [35,36]. The association between circadian misalignment and cognitive impairment has been studied [37,38], and an evidence of a mechanism involving the molecular clock has been presented [39]. The suprachiasmatic nucleus degenerates, and melatonin is not secreted rhythmically due to changes in the expression of circadian clock genes, such as Bmal1, inducing accumulation of amyloid plaques that cause dementia [40,41]. In addition, those who cannot manage their work hours may become stressed. Glucocorticoid hormones secreted during increased stress affects the regulation of the glucocorticoid receptors in the hippocampus, which is responsible for learning and memory function [42]. The increased stress may weaken the regulation of negative feedback of the hypothalamic-pituitary-adrenocortical axis through the downward regulation of hippocampal corticosteroid receptors.

This problem is beyond the knowledge and decision-making capacity of individual workers because workers who lack job control and job authority cannot control their working hours. In addition, it can be an important issue not only for those who work for long hours but also for those who work for normal hours. In relation to the regularity of working hours, we hope that our findings will be helpful to support policy changes regarding workers’ health by preparing a response system under various working conditions.

## 5. Limitations

Our study has some limitations. First, our study assessed cognitive function using MMSE alone. There are other assessment tools for cognitive function [43,44], with the Montreal Cognitive Assessment being widely used [45]. Although MMSE is widely used and its validity has been studied [14], it would have helped if data on history of behavioral and cognitive changes that are important to diagnose cognitive impairment were available. Second, our study did not consider the occupations and working environments of the workers. The emotional response or vulnerability of workers vary depending on their occupations and this may have a different effect on cognitive function [46]. Further, the working environment could be subdivided considering information on whether the shift work was permanent or rotating and the degree of job control. However, our sample size was representative of the middle-aged and older population of South Korea. In addition, a few prospective studies have focused on the regularity of working hours rather than shift work or long working hours, and such studies allow an intra-individual approach that controls many confounders studied in other research.

## 6. Conclusions

In conclusion, our study documented that the regularity of working hours was associated with cognitive impairment. In the current situation, where the population of middle-aged and older workers is increasing, we were able to confirm the effects of irregular work schedules on health. In addition, our findings are also relevant for workers who maintain the legal working hours, and thus, a better work schedule that can maintain the regularity of working hours should be structured for middle-aged workers in the near future.

## Figures and Tables

**Figure 1 ijerph-19-04161-f001:**
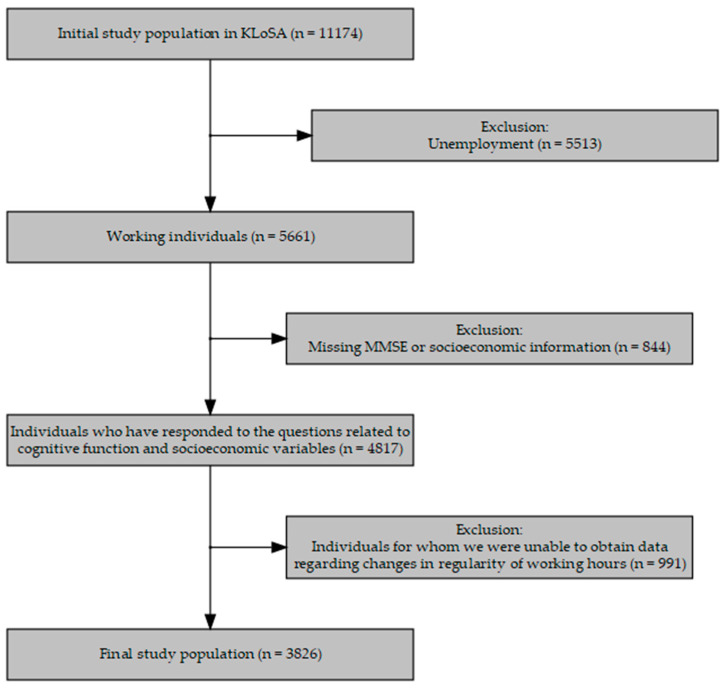
Flowchart showing the selection of the study participants. KLoSA, Korean Longitudinal Study of Aging; MMSE, mini–mental state examination.

**Figure 2 ijerph-19-04161-f002:**
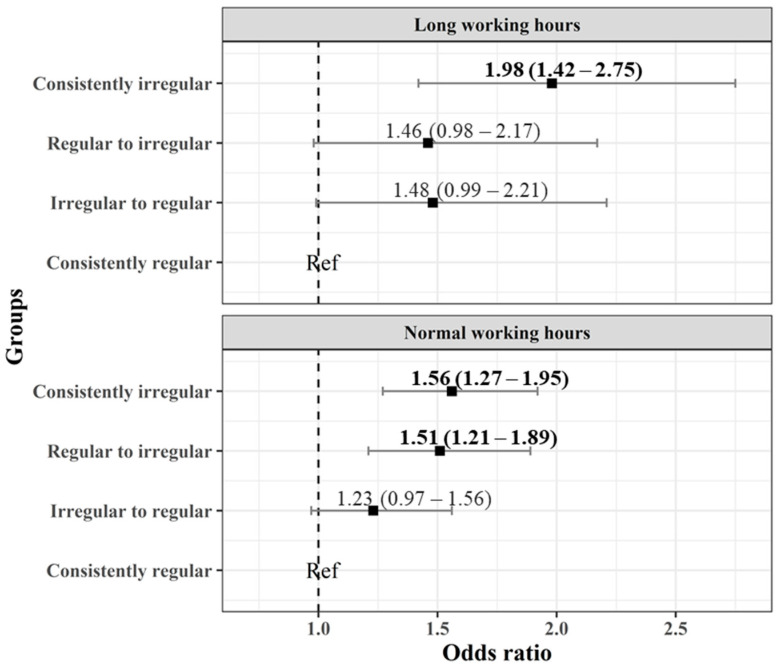
Odds ratio of cognitive impairment according to changes in the regularity of working hours with long and normal working hours (“consistently regular” group being a reference).

**Table 1 ijerph-19-04161-t001:** General characteristics and cognitive impairment status of the study population at baseline.

	Total	Noncognitive Impairment	Cognitive Impairment	*p*-Value
	(*n* = 3826)	(*n* = 3361)	(*n* = 465)	
MMSE				
Mean (SD)	27.2 (± 3.42)	28.2 (± 1.83)	19.9 (± 3.54)	<0.001
Work regularity (baseline)				
Regular	2029 (53.0%)	1862 (55.4%)	167 (35.9%)	<0.001
Irregular	1797 (47.0%)	1499 (44.6%)	298 (64.1%)	
Changes in work regularity				
Consistently irregular	1215 (31.8%)	979 (29.1%)	236 (50.8%)	<0.001
Regular to irregular	531 (13.9%)	462 (13.7%)	69 (14.8%)	
Irregular to regular	582 (15.2%)	520 (15.5%)	62 (13.3%)	
Consistently regular	1498 (39.2%)	1400 (41.7%)	98 (21.1%)	
Sex				
Male	2226 (58.2%)	2031 (60.4%)	195 (41.9%)	<0.001
Female	1600 (41.8%)	1330 (39.6%)	270 (58.1%)	
Age, years				
45–54	1307 (34.2%)	1246 (37.1%)	61 (13.1%)	<0.001
55–64	1464 (38.3%)	1341 (39.9%)	123 (26.5%)	
65–74	823 (21.5%)	644 (19.2%)	179 (38.5%)	
75 or above	232 (6.1%)	130 (3.9%)	102 (21.9%)	
Income				
1Q	918 (24.0%)	662 (19.7%)	256 (55.1%)	<0.001
2Q	985 (25.7%)	879 (26.2%)	106 (22.8%)	
3Q	963 (25.2%)	894 (26.6%)	69 (14.8%)	
4Q	960 (25.1%)	926 (27.6%)	34 (7.3%)	
Education level				
Elementary school	1257 (32.9%)	925 (27.5%)	332 (71.4%)	<0.001
Middle school	718 (18.8%)	652 (19.4%)	66 (14.2%)	
High school	1359 (35.5%)	1304 (38.8%)	55 (11.8%)	
University	492 (12.9%)	480 (14.3%)	12 (2.6%)	
Residential area				
Urban	1501 (39.2%)	1384 (41.2%)	117 (25.2%)	<0.001
Small city	1223 (32.0%)	1107 (32.9%)	116 (24.9%)	
Rural	1102 (28.8%)	870 (25.9%)	232 (49.9%)	
Marital status				
Married	3319 (86.7%)	2970 (88.4%)	349 (75.1%)	<0.001
Separated/Not married	507 (13.3%)	391 (11.6%)	116 (24.9%)	
Smoking				
Current smoker	960 (25.1%)	882 (26.2%)	78 (16.8%)	<0.001
Ex-smoker	619 (16.2%)	561 (16.7%)	58 (12.5%)	
Non-smoker	2247 (58.7%)	1918 (57.1%)	329 (70.8%)	
Drinking				
Current drinker	1964 (51.3%)	1783 (53.0%)	181 (38.9%)	<0.001
Ex-drinker	368 (9.6%)	320 (9.5%)	48 (10.3%)	
Non-smoker	1494 (39.0%)	1258 (37.4%)	236 (50.8%)	
Working hours				
Long	1153 (30.1%)	1044 (31.1%)	109 (23.4%)	<0.001
Normal	2673 (69.9%)	2317 (68.9%)	356 (76.6%)	
Chronic disease *				
Yes	1409 (36.8%)	1194 (35.5%)	215 (46.2%)	<0.001
No	2417 (63.2%)	2167 (64.5%)	250 (53.8%)	
Social activity				
Active	3155 (82.5%)	2862 (85.2%)	293 (63.0%)	<0.001
Inactive	671 (17.5%)	499 (14.8%)	172 (37.0%)	

MMSE, Mini–Mental State Examination; SD, standard deviation. * Chronic diseases include hypertension, diabetes, malignancy, chronic lung disease, liver disease, heart disease, and cerebrovascular disease.

**Table 2 ijerph-19-04161-t002:** Results of the generalized estimating equation model on cognitive impairment.

Variable	ORs *	95% CI
Regularity		
Consistently regular	1.00	Ref
Consistently irregular	**1.56**	**(1.30–1.88)**
Regular to irregular	**1.46**	**(1.20–1.77)**
Irregular to regular	**1.24**	**(1.01–1.53)**
Sex		
Male	1.00	Ref
Female	1.39	(1.11–1.74)
Age, years		
45–54	1.00	Ref
55–64	1.34	(1.06–1.70)
65–74	2.38	(1.82–3.11)
75 or above	3.9	(2.84–5.34)
Income		
1Q	1.36	(1.07–1.73)
2Q	0.95	(0.75–1.21)
3Q	1.06	(0.84–1.33)
4Q	1.00	Ref
Education level		
Elementary school	4.17	(2.96–5.88)
Middle school	2.4	(1.68–3.42)
High school	1.51	(1.08–2.12)
University	1.00	Ref
Residential area		
Urban	1.00	Ref
Small city	1.22	(1.00–1.49)
Rural	1.36	(1.12–1.65)
Marital status		
Married	1.00	Ref
Separated/Not married	1.23	(1.02–1.48)
Smoking		
Current smoker	0.96	(0.75–1.22)
Ex-smoker	1.00	(0.78–1.28)
Non-smoker	1.00	Ref
Drinking		
Current drinker	1.18	(0.94–1.47)
Ex-drinker	0.99	(0.83–1.18)
Non-smoker	1.00	Ref
Working hours		
Long	0.94	(0.80–1.09)
Normal	1.00	Ref
Chronic disease		
Yes	1.11	(0.96–1.28)
No	1.00	Ref
Social activity		
Active	1.00	Ref
Inactive	1.98	(1.70–2.30)

* Fully adjusted model; ORs, odds ratio; CI, confidence interval.

## Data Availability

The datasets analyzed during this study are available in the KLoSA repository, https://survey.keis.or.kr/eng/klosa/klosa01.jsp (accessed on 12 May 2021).

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
