# Peer review of "Association between Changes in the Regularity of Working Hours and Cognitive Impairment in Middle-Aged and Older Korean Workers: The Korean Longitudinal Study of Aging, 2008–2018"

_ijerph, 2022, doi:10.3390/ijerph19074161_

Round 1

Reviewer 1 Report

The reviewer has a couple of questions:

  1. in line 71, what is systematic sampling?
  2. in line 71, data were from 2006, 2008, 2010, 2012, 2014, 2016, and 2018. the data were published every two years?
  3. in lines 86-89, about the 2-year lag-time, how to understand "Thus, we categorized the changes 87 into “consistently irregular group,” “consistently regular group,” “regular to irregular 88 group,” and “irregular to regular group” through the gap of the 2 years."

Author Response

Responses to the comments of Reviewer #1

Thank you for taking the time to read our original manuscript and to provide feedback. We have enclosed a point-by-point list of changes or rebuttals along with our revised manuscript.

Reviewer Point P 1 – in line 71, what is systematic sampling?

(Response) Thank you for your detailed comments. When the Korea Employment Information Service (KEIS) extracted elderly households in Korea by multistage, stratified design, sampling was conducted at regular intervals. Perhaps the meaning was misrepresented while receiving English proofreading. We edited our methods section to make it easier for readers to understand as your comment.

Line 67 “In 2006, the KEIS extracted the data through systematic sampling, using a multistage design of stratification by region and residential type, and provided socioeconomic and health data of middle-aged people aged ≥45 years residing in Korea.”

Reviewer Point P 2 – in line 71, data were from 2006, 2008, 2010, 2012, 2014, 2016, and 2018. the data were published every two years? in lines 86-89, about the 2-year lag-time, how to understand "Thus, we categorized the changes into “consistently irregular group,” “consistently regular group,” “regular to irregular 88 group,” and “irregular to regular group” through the gap of the 2 years."

(Response) We apologize for making it difficult to convey the meaning of the sentence. As the reviewer understood, these data were collected and published every two years. Therefore, the change in the regularity of working hours that we are interested in changes every two years from the second follow-up, and can be classified into four categories “consistently irregular group,” “consistently regular group,” “regular to irregular group,” and “irregular to regular group”. We revised the sentence as follows.

Line 91 “From the second follow-up, we categorized the changes into “consistently irregular group,” “consistently regular group,” “regular to irregular group,” and “irregular to regular group” through the gap of the 2 years.”

Reviewer 2 Report

The manuscript “Association between Changes in the Regularity of Working Hours and Cognitive Impairment in Middle-aged and Older Korean Workers: The Korean Longitudinal Study of Aging,

2008–2018” describes a study aiming at analyze whether cognitive decline occurs more frequently in middle-aged and elderly workers due to changes in work regularity and the association of cognitive decline and working hours.

The question posed by the authors is interesting and appears to fit the aims and scope of the journal. The design of the study is clear and the statistical analyses are appropriate to the study design. I think that the article should be considered for publication after some minor revisions. Therefore, I would suggest to the Authors some clarifications in order to improve the overall quality of the study.

- The authors should highlight in Table 2 and Figure 2 which ORs are significant (e.g. reporting values in bold).

- The sentence at lines 168-170 “On the other hand, participants in the “regular to irregular” subgroup were only related to the group with normal working hours” is not clear. Please reformulate.

- The content of the sentence at lines 191-192 “When analyzed in the subgroup, working hours were associated with the deterioration of cognitive function” should be deepened.

- The sentence at lines 192-194 “We noted an unexpected result among workers who maintained the statutory working hours: the risk of cognitive deterioration was high in those who worked with a worsened regularity” is not clear. The authors should explain why this result was unexpected.

- The conclusive sentence is confusing. The results of the study showed that not only the changes in the regularity of working hours were associated with cognitive impairment but also the consistently irregular condition. Please reformulate.

Author Response

Responses to the comments of Reviewer #2

Thank you for taking the time to read our original manuscript and to provide feedback. We have enclosed a point-by-point list of changes or rebuttals along with our revised manuscript.

Reviewer Point P 1 – The authors should highlight in Table 2 and Figure 2 which ORs are significant (e.g. reporting values in bold).

(Response) Thank you for your detailed comments. We have modified important parts of Table 2 and Figure 2 in bold type and indicated as follows.

Line 155

Table 2. Results of the generalized estimating equation model on cognitive impairment.

Variable

ORs*

95% CI

Regularity

Consistently regular

1.00

Ref

Consistently irregular

1.56

(1.30–1.88)

Regular to irregular

1.46

(1.20–1.77)

Irregular to regular

1.24

(1.01–1.53)

Line

Reviewer Point P 2 – The sentence at lines 168-170 “On the other hand, participants in the “regular to irregular” subgroup were only related to the group with normal working hours” is not clear. Please reformulate.

(Response) Thank you very much for pointing this out. We considered what is the appropriate sentence for this context again. We revised this sentence as follows:

Line 168 “Supplementarily, "regular to irregular" subgroup workers had large ORs values regardless of working hours, but was only significant in the group with normal working hours.”

Reviewer Point P 3&4 – The content of the sentence at lines 191-192 “When analyzed in the subgroup, working hours were associated with the deterioration of cognitive function” should be deepened. The sentence at lines 192-194 “We noted an unexpected result among workers who maintained the statutory working hours: the risk of cognitive deterioration was high in those who worked with a worsened regularity” is not clear. The authors should explain why this result was unexpected.

(Response) Thank you for your constructive comment. We edited and rearranged discussions on subgroup analysis related to working hours. We expected that workers with long working hours would be more vulnerable to regularity of working hours. However, this was not the case in "regular to irregular” group. And we edited the sentence to make it easier to understand as follows:

Line 191 “ In the case of working hours analyzed by subgroup, the working hours were found to affect the deterioration of cognitive function. A study on middle-aged people using the same data found that women with long working hours had less cognitive ability after 5 years [27].  We expected that the effect of working hour regularity would be greater in one of the subgroups of working hours. In the "consistently irregular" group, workers with long working hours had a higher risk of cognitive decline, but in the "regular to irregular" group, only workers with statutory working hours were significant.”

 Reviewer Point P 5 – The conclusive sentence is confusing. The results of the study showed that not only the changes in the regularity of working hours were associated with cognitive impairment but also the consistently irregular condition. Please reformulate.

(Response) We agreed with the reviewer on this point. The 241st line of "changes in the regularity of working hours" can confuse that only "regular to irregular" condition can affect. Therefore, we can summarize this study by omitting the word "change", and using the expression effects of irregular work schedules in the next sentence. We edited conclusion part as follows:

Line 241 “In conclusion, our study documented that the regularity of working hours was associated with cognitive impairment.”

Line 244 “In addition, our findings are also relevant for workers who maintain the legal working hours, and thus, a better work schedule that can maintain the regularity of working hours should be structured for middle-aged workers in the near future.”

Reviewer 3 Report

The aim is interesting. However, there are two critical issues (cognitive impairment concept/scores, and assessment). It will be important for the authors to clarify the concept of cognitive impairment when it comes to severity (mild, major) and the associated MMSE scores (a literature review is suggested), adjusted by age and educational level. Some statements may confuse readers (e.g., “The impairment, represented by Alzheimer´s disease …”). It was not clear why cognitive impairment was associated with a specific disease, when in table 1 there is no clinical diagnosis. Assessment the cognitive function with only one screening instrument is not recommended and may become inconclusive.

Author Response

Responses to the comments of Reviewer #3

Thank you for taking the time to read our original manuscript and to provide feedback. We have enclosed a point-by-point list of changes or rebuttals along with our revised manuscript.

Reviewer Point P 1 – It will be important for the authors to clarify the concept of cognitive impairment when it comes to severity (mild, major) and the associated MMSE scores (a literature review is suggested), adjusted by age and educational level.

(Response) Thank you for your constructive comment. There have been studies suggesting that different cut-off values for MMSE should be applied according to age or educational level. However, although there is no great difficulty in performing screening by applying this to individuals, it was difficult for the authors to adjust the result value in analysis studies, and one cut-off value was used in other studies. The authors also understood this problem and stated that we could not analyze our study by applying different cut-off values to individuals.

Line 80 “Since the MMSE score varies according to age and educational level, there are several normative studies that analyze each cutoff value [15,16]. However, MMSE is not a diagnostic test, but a screening function that enables disease suspension, and there is a domestic analysis suggesting that it is reasonable to set the cut-off of the MMSE for the age of the subject in our study in a traditional way [17].”

Reviewer Point P 2 – Some statements may confuse readers (e.g., “The impairment, represented by Alzheimer´s disease …”). It was not clear why cognitive impairment was associated with a specific disease, when in table 1 there is no clinical diagnosis.

(Response) Thank you for your detailed comments. The statement for Alzheimer's disease, which you mentioned, was deleted because it seemed inappropriate. In addition, it was not initially included that cognitive impairment is related to various health conditions and chronic diseases.  In this regard, it was revised by adding that several chronic diseases can affect mild cognitive impairment.

Line 39 “The impairment is common in the elderly population, and the resulting burden of healthcare costs is high [7,8].”

Line 100 “In addition, there have been studies showing that chronic diseases, particularly diabetes, hypertension, and chronic kidney disease, are associated with dementia or mild cognitive impairment in the elderly [20-22].”

Reviewer Point P 3 – Assessment the cognitive function with only one screening instrument is not recommended and may become inconclusive.

(Response) We agreed with the reviewer on this important point. Since KLoSA is a sample collected to see various results according to aging, it was not possible to collect various tests and questionnaires for cognitive function. As we wrote in our limitation, the cognitive function needs to proceed with the history taking while interviewing the doctor. We thought it would be a better study if there were other instruments, but we regret that this is not possible in this data.

Round 2

Reviewer 3 Report

Thanks for your attention to my comments. I understand the difficulties in overcoming the two critical issues. However, the changes made are positive.